# Elephants Not in the Room: Systematic Review Shows Major Geographic Publication Bias in African Elephant Ecological Research

Rachael B. Gross  and Robert Heinsohn *

Fenner School of Environment and Society, Australian National University, Canberra, ACT 2601, Australia; rachael.gross@anu.edu.au
* Correspondence: robert.heinsohn@anu.edu.au

**Abstract:** African savanna elephants (*Loxodonta africana*) are a keystone species in African ecosystems. As a result of increasing anthropogenic pressure, elephant populations have declined significantly in the last two centuries. Research on a broad sample of these populations is necessary to inform management strategies over a range of environmental and socio-political conditions. In order to evaluate the current state of literature that is informing evidence-based management and conservation of elephants, we systematically reviewed all research published on the ecology of African elephants from the last 20 years (492 publications). We contrasted the geographic distribution of published research against the 2016 IUCN elephant census. We found several statistically significant biases in the geographic distribution of elephant research. South Africa has 4.54% of the total elephant population and accounted for 28.28% of all research publications. Kenya has 5.49% the total elephant population but accounts for 20.6% of the research. Conversely, Botswana has 31.68% of the total elephant population but accounts for only 9.29% of the research and Zimbabwe has 19.89% of the total elephant population with only 10.50% of research. We also found that 41.85% of areas with ~60,100 elephants have not had any research published on their populations in the last 20 years. This publication imbalance may encourage management strategies that are overly dependent on misrepresentative information from a small subset of the elephant population. We recommend that (1) marginalised areas with large elephant populations (e.g., Botswana and Zimbabwe) should receive higher priority for future research, (2) new research and proposals should design theoretical frameworks to account for and overcome the present biases, and (3) local community-based management approaches should be prioritised and amplified in order to overcome the barriers to conducting research in priority areas.

**Keywords:** research bias; African elephant conservation; *Loxodonta africana*; community-based management; conservation; policy



## 1. Introduction

The conservation and management of African savanna elephants (*Loxodonta africana*, hereafter elephant) has been a focus of research for decades. Elephants play a significant role in maintaining and shaping ecosystems across their distribution in Africa [1]. They are a major keystone and umbrella species meaning that their activities affect other wildlife, many of which are endangered [1,2]. Elephant meta-populations in Africa, including the forest elephants (*Loxodonta cyclotis*), have declined significantly since, and due to, European colonisation throughout the 19th century. African savanna elephants are listed as endangered and forest elephants as critically endangered by the International Union of the Conservation of Nature (IUCN) [3,4]. Declines in elephant meta-populations are largely attributed to colonial anthropogenic factors, including hunting for the illegal ivory trade and conflict with humans [4]. Climatic changes, including increasing surface temperatures and more severe drought conditions, are also increasingly prevalent stressors for elephants [5–16].

Currently, 84% of African savanna elephants reside within Protected Areas [PAs] [4]. Fenced areas generally experience population increases because wildlife are more protected from negative human interactions [17], although PAs also have high levels of elephant mortality linked to illegal killing [4,17]. Elephants can attract much needed revenue, with wildlife tourism garnering US$38 billion across Africa in 2018 and attracting more than 67 million international tourists across the continent [18]. While most PAs are not fenced many that are report elephant populations that are reaching and, in some cases breaching, their contested [19–24] carrying capacity [25–27].

Landscapes that have elephant populations that are over carrying capacity can undergo major ecosystem change [3,25,28,29]. Elephants can cause permanent changes to old growth trees and riparian systems and inhibit ecosystem regeneration [30–37]. These effects can largely be attributed to their habitat use and foraging habits [5,38]. They also compete with many species of mega- and mesoherbivores for these resources [39–41]. Elephant overpopulation is a landscape scale conservation problem. Many fenced PAs, which are generally small patches of fragmented landscape, are subject to overpopulation by elephants [27] and are experiencing difficulty in maintaining healthy ecosystems with managers struggling to control populations ethically and within the public eye [27]. The ecosystem change is exacerbated by the fact that PAs are often home to other vulnerable and endangered species that are at risk from elephant induced habitat change [2]. Outside of fenced PAs, elephant populations continue to decline due to lack of protection and increasing human/elephant conflict [4,42–44]. Their decline in outside areas could lead to shifts in ecosystem function and quality and disrupt the ecological function and maintenance of landscapes outside of PAs.

The conservation of elephants is critical from both an economic and environmental perspective but adding the uncertainty of the future under the effects of climate change means there is a critical need to fully understand the remaining meta-population to inform the best management strategies appropriate for each region. It is becoming evident that current strategies to manage elephants are economically and ecologically inefficient, largely due to the increasing pace of environmental change and chronic underfunding and lack of resources in the conservation sector [45].

Peer-reviewed research is a key input into conservation policy and management [46]. However, recent studies have shown a bias in research on conservation management strategy [47,48]. A study has also recently shown that knowing the current state of research for conservation is essential for effectively coordinating future research efforts [46]. Basing policy and management on information that is not representative of a specific context may lead to inappropriate or ineffective strategies [49].

The purpose of this paper is to evaluate whether the recent scientific research on elephants available to managers and conservationists is geographically diverse enough to inform managers across the species' range. We aimed to identify gaps and biases in research on elephants, determine the implications of any research biases, and make recommendations for future research. We systematically reviewed all peer-reviewed published material on elephants from the last 20 years (492 papers) and contrasted the results against the 2016 IUCN Elephant Census. Our hypotheses were as follows. First, there will be a geographical bias in published research towards countries with established elephant research units/non-government organisations and better accessibility to elephants, such as South Africa, Kenya and Botswana. Second, there will be a large proportion of the elephant meta-population that has not been studied recently, for example in less politically stable countries such as Zimbabwe that have pre-existing systemic barriers to research such as funding and administrative limitations.

## 2. Materials and Methods

### 2.1. Data Collection

We systematically reviewed all available peer-reviewed research on elephants over 20.5 years from 2000 until mid-2020. Search engines used were Google Scholar, Scopus, Web

of Science, and the Australian National University Library Supersearch Engine. We filtered by date range (1 January 2000–30 July 2020) and article type. Media and creative pieces were omitted because they can be speculative and subjective. To ensure we reviewed all relevant literature, we did not use refining search terms like "ecology" so as not to exclude any work and reviewed all research pertaining to elephants and refined the data upon completion. We used the search terms "African elephant", "*Loxodonta africana*", "African savanna elephant", "savanna elephant", "Loxodonta" and "*africana*". We defined ecology as any interaction or relation to elephants and their environment as this is the main factor that will shape and change elephant behaviour and movement in the future. Research on basic biology and non-environmental biochemistry, physiology and captive specimens was not recorded as it is feasible for these topics to be extrapolated to a wider population based on a single or captive population. For each relevant article, we recorded the title, authors, year of publishing, country/countries, whether the area was partially, fully fenced or unfenced, whether the area was transfrontier, if the area is privately or publicly owned and/or managed and the name of the area or reserve. We also recorded the topic of the research within five categories that we believe represent the most pertinent issues in elephant conservation. Definitions of these categories can be found in Table 1. We have used three categories for some topics as most options are not simply binary (e.g., fenced or not fenced). If the data for a category was not available, they were excluded from specific calculations but included in the overall calculations. For example, if one of the studies took place in a park where the fencing status is not clear, it was omitted from the analysis relating to fencing but included in the overall review based on country and PA. If a paper covered more than one topic, then it was recorded under both topics. We reviewed research published in all languages assuming non-English papers would appear by using the scientific name, and variations (aforementioned), as a search term.

**Table 1.** Definitions of each category used to refine this systematic review.

| Category | Sub-Category | Definition |
|---|---|---|
| Fencing | Fully fenced | The area is fully enclosed by a fence capable of confining elephant movement |
| | Partially fenced | The area is partially enclosed by either wildlife management or agricultural fencing or contains internally fenced areas |
| | Unfenced | The area is unfenced or lacks fencing adequate to inhibit elephant movement |
| Frontier | Transfrontier | The area intersects several countries and nations |
| | National frontier | The area is fully within the boundaries of one country |
| Management | Government | The area is under the funding and management of the government or a governmental body/ies, includes community managed areas |
| | Private | The area is privately owned and managed |
| | Joint | The area is managed by both the government/community and privately |
| | Ecology | The article is focused on how elephants interact with or impact the environment around them |
| | Behaviour | The article is focused on how and why elephants behave in certain ways and how those behaviours have changed |
| | Climate interactions | The article is focussed on how the varying aspects of the climate interact with elephant survival and livelihood |
| | Illegal Killing | The article is focused on the impact of, history or predictions of illegal killing of elephants, including for both commercial savanna meat and ivory |

We did not record work on the African forest elephant (*Loxodonta cyclotis*) as research on that species is limited due to high risk from political unrest in the countries they reside within [50,51]. We included countries with a crossover zone between the species such as Uganda but excluded parks with the *L. cyclotis* species and to our knowledge there are no PAs with both species present. We limited the search to the last 20 years because the main interest was understanding how elephants need to be managed in changing environmental conditions coinciding with the major topics we investigated (Table 1) where research preceding this time period may be less relevant to contemporary elephant ecology [52–54].

Unimpeded elephant populations can also double naturally within 10 years [55,56] and the illegal killing rate/ivory trade fluctuates annually so evidence from elephant populations that is more than 20 years old may not be comparable to modern studies or as relevant for policy formation.

We then reviewed the 2016 International Union for the Conservation of Nature (IUCN) Elephant Status Report [3], which included a continent-wide elephant meta-population survey from 2015. We believe this report to be the most reliable and recent dataset on meta-populations of elephants across Africa. The report is the first and latest continent-wide survey of its kind. From this report, we extracted the total estimate across all countries as well as meta-population estimates for each country and for each area surveyed. We used the estimate of 415,428 as the total number of individuals in the elephant meta-population. We grouped a set of small PAs under "Other Private Reserves". These comprise a series of reserves bordering Kruger National Park in South Africa that were grouped together in the Census (Maremani, Kwandwe, Kariega and Balule, Timbavati, Umbabat and Klaserie).

*2.2. Data Analysis*

We calculated the proportion of African elephants that resided within each country as a percentage, and used the estimates provided by the report. We calculated the results against the provided upper and lower limits of the confidence interval to ensure there were no notable discrepancies or outliers influencing the results.

We then took the total number of papers published per country and per area/reserve and calculated it as a percentage of the total. We contrasted the available list of areas surveyed by the census against the list of reserves where research had been conducted to indicate which places had not had research done.

To test if there were statistically significant biases present between countries, we standardised the meta-population to 100% to exclude the *L. cyclotis* species and calculated the number of papers that should have been published for each country if it had been based on that country's proportion of the elephant meta-population. We conducted Chi-Squared tests on this data, omitting Eritrea as its elephant population is extinct. This process was then repeated with data related to the specific PAs, the fencing status of PAs and the custodianship of land which elephants inhabit. A data summary is available in the Supplementary Materials.

**3. Results**

We recorded over 8000 published articles and analysed 492 that were considered relevant under our criteria. There were 14 papers published in areas not surveyed in the census which are not counted in the results on biases in PAs but counted in the analysis by country. There were 28 papers (5.69%) that took place in multiple parks/countries and 4 papers (0.81%) that were published in transfrontier parks that were counted in all of the analyses where applicable. There were four papers recorded that were published in languages other than English, two in French and two were in Portuguese. A list of the articles included in this analysis can be provided on request.

*3.1. Distribution of Published Research*

The countries with the highest proportion of research on elephants published in the last 20 years were South Africa (28.3%) and Kenya (20.6%); these had, respectively, 4.5- and 5.5% of the total meta-population (Figure 1). The countries with the highest elephant numbers were Botswana (31.7% of total) and Zimbabwe (19.9%) and these published 9.3% and 10.5% of research respectively. There was a highly significant statistical difference between the actual number of papers published and the number that would have been published in each country if based on its proportion of the total African elephant meta-population ($\chi^2$ = 964.55, $p$ < 0.001, 17 d.f.).

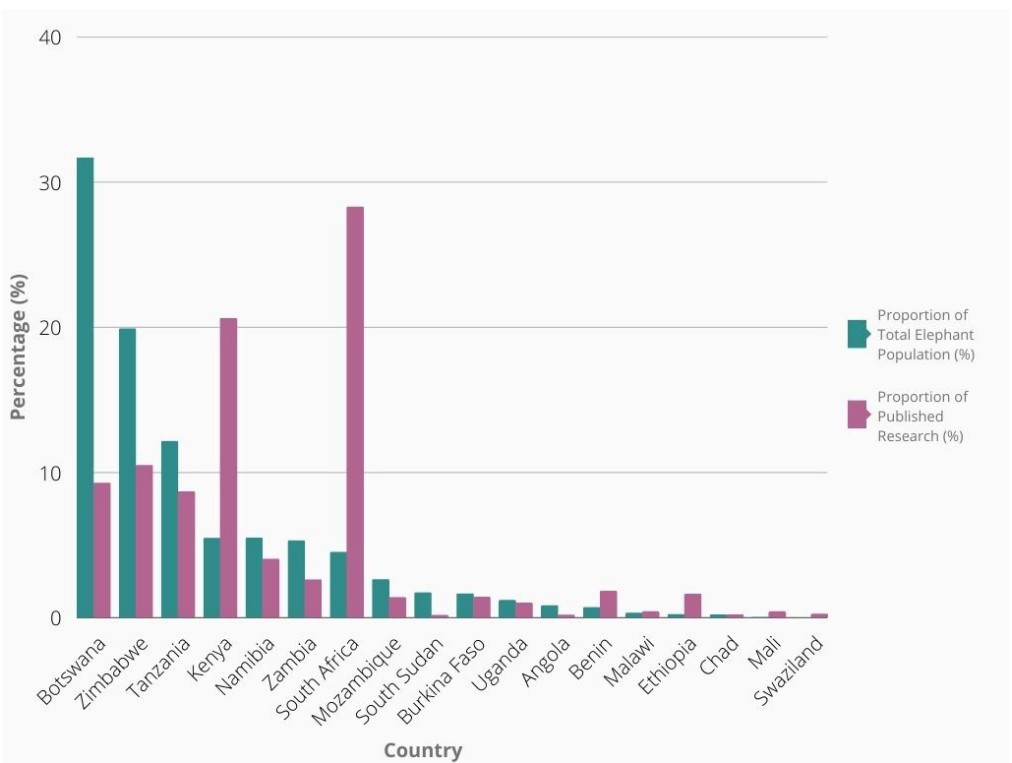

**Figure 1.** Research and populations by country.

*3.2. Research and Populations by Protected Areas*

3.2.1. Highest Proportion of Research

The PAs with the most research on elephants published over the last 20 years were Kruger National Park (12.2%) and Other Private Reserves (9.7%) which are all located around Kruger National Park. Both areas are in South Africa and have 4.5% and 0.6%, respectively of all elephants (Figure 2). Addo Elephant Park, Pilanesburg Provinicial Reserve and Tembe Elephant Park are also in South Africa and all show a large proportion of research given their elephant population sizes. As in Figure 1, Kenya is also overrepresented in the research as the Laipikia-Samburu and Amboseli Ecosystems have considerable research and small elephant populations. There was a highly significant statistical difference between the actual number of papers published and the number that would have been published in each PA based on its proportion of the total African elephant meta-population ($\chi^2$ = 2564.03, *p* < 0.001, 12 d.f.).

3.2.2. Highest Proportion of the Elephant Population

The PAs with the highest elephant populations are Northern Botswana (31.3%) and Hwange National Park in Zimbabwe (11%) and have produced 8.4% and 5.3% of elephant research over the last 20 years (Figure 2). Other PAs underrepresented in the research are the Ruaha-Runga Ecosystem in Tanzania and the Zambezi Region in Zambia. There was a highly significant statistical difference between the actual number of papers published and the number that would have been published in each PA had it been based on proportion of the total African elephant meta-population ($\chi^2$ = 3828.44 *p* < 0.001, 51 d.f.).

3.2.3. PAs with and without Research

Countries with small elephant populations like South Africa and Namibia have small percentages of the elephants that have not been studied. Countries with high populations including Zimbabwe, Mozambique and Angola, show up to 80% of PAs with elephants that have not had research published about them over the last 20 years (Figure 3). Countries with low populations and high political conflict, especially Senegal, Niger and Eritrea, have

the highest percentage of population not studied (Figure 3). These countries, plus the next most understudied places, are all within Central and Northern Africa.

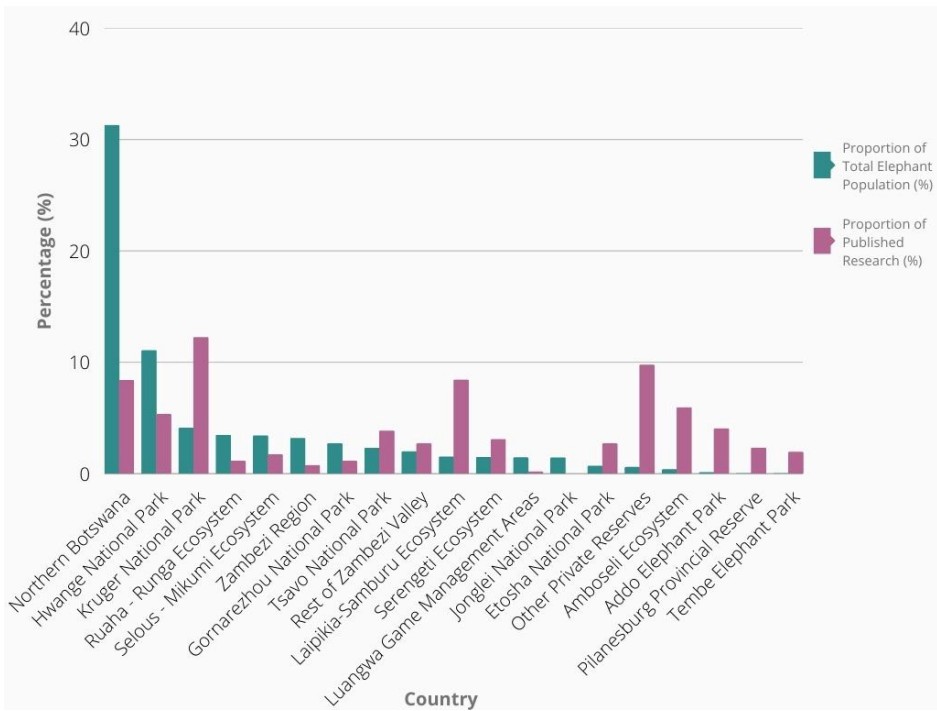

**Figure 2.** Research and populations by National Parks & Reserves and Protected Areas (NPR/PAs).

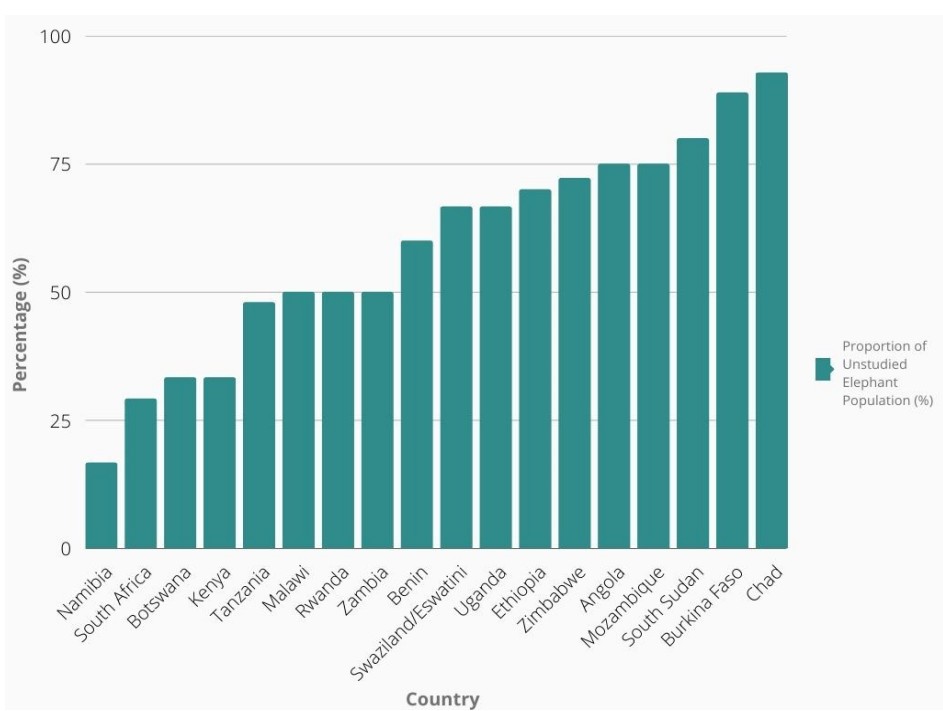

**Figure 3.** Percentage of PAs by country where no research on elephants has been published over the last 20 years.

In the last 20 years, 40.8% of PAs with elephants have not had any research published. This equates to ~68,500 elephants that have not been studied by researchers over the last 20 years.

### 3.2.4. Fencing and Custodianship of Elephant Habitats

A large proportion (43.3%) of research on elephants in the last 20 years took place in unfenced PAs and reserves, followed by partially fenced (30.6%) and fenced PAs [26.1%]. Comparatively, 7.5% of elephants reside in fenced reserves, 64.4% in partially fenced PAs and 28.2% in unfenced PAs (Figure 4). There was a highly significant statistical difference between the actual number of papers published and the number that would have been published in fenced, partially fenced, and unfenced PAs if based on the proportion in each of the total African elephant meta-population ($\chi^2$ = 324.65, *p* < 0.001, 2 d.f.).

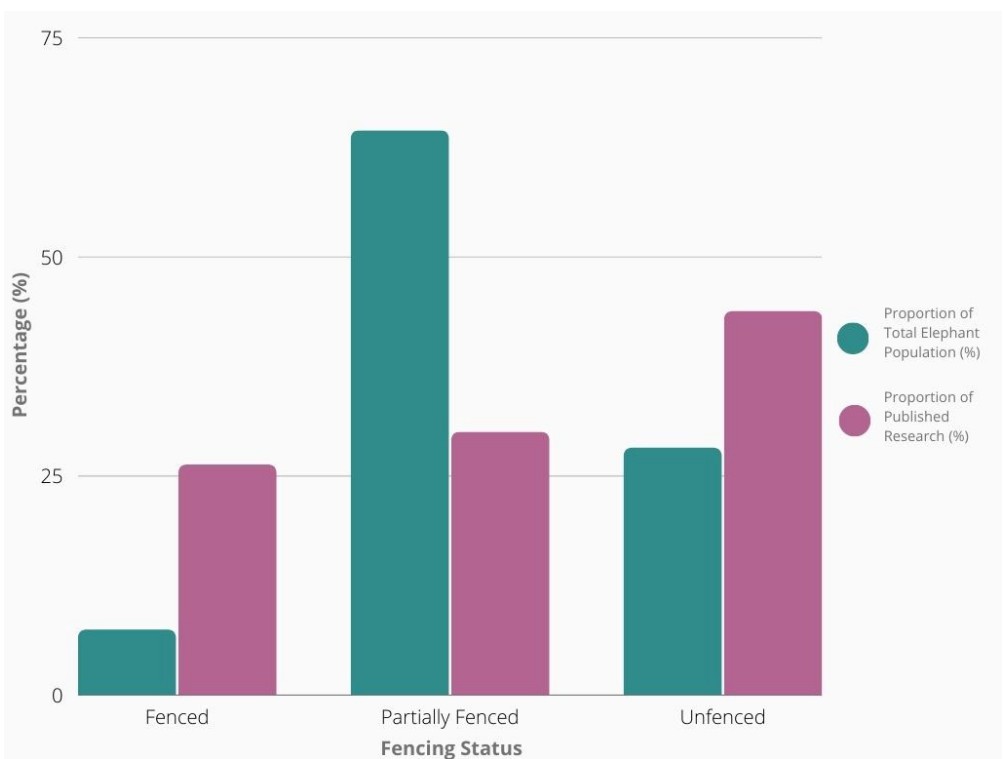

**Figure 4.** Percentage of research versus proportion of elephant population occurring in fenced, partially fenced and unfenced reserves (PAs).

The majority (52.6%) of research on elephants in the last 20 years took place in PAs publicly owned and managed by the government/state, followed by co-owned and managed PAs (30.4%) and privately-owned reserves (17%). Most research on elephants has occurred in government-managed PAs that account for 6.1% of the total elephant population. Comparatively, jointly managed PAs have 65% of the total elephant meta-populations, followed by publicly managed PAs with 28.9% of the meta-population with private reserves managing 6% of the total meta-population. There was a highly significant statistical difference between the actual number of papers published and the number that would have been published in each category of custodianship if based on the proportion in each of the total African elephant meta-population ($\chi^2$ = 268, *p* < 0.001, 2 d.f.).

### 3.3. Research Topics

The majority of research (41.5% and 30.1% consecutively) on elephants has been on the topics of Ecology and Behaviour in the last 20 years. Human/Elephant conflict is the third most common topic at 15.7%, followed by Climate (7.1%) and Poaching (5.7%). The research topic percentage in each country can be found in the Supplementary Materials.

## 4. Discussion

Published research on African elephants over the last 20 years has shown potentially concerning geographical and population bias. We define geographic bias as more published research coming from certain countries and regions, and population bias as more published research coming from certain PAs within those countries. A significant geographical bias is evident as more research on elephants is being published in certain countries, predominantly South Africa and Kenya, where elephant populations represent a smaller proportion of the total across Africa than other countries such as Botswana and Zimbabwe (Figure 1).

Further, a significant population bias is present in favour of elephant populations within parks such as Kruger National Park, Associated Private Reserves (both in South Africa) and Amboseli National Park (Kenya) (Figure 2). Researchers on these parks have produced a large amount of published research on elephants compared to the size of the elephant populations (Figure 2). In line with the geographic bias, much larger populations of elephants, for example in Northern Botswana and Hwange National Park (Zimbabwe), have less research published in proportion to their size (Figure 2). Further the geographic and population bias in elephant research reported here also shows that a large proportion of elephants have not been studied at all over the last 20 years. Most prominently, 40.8% of the PAs have not had any research published on their elephant populations over this period. The population bias is strongly supported by some countries [South Sudan, Burkina Faso, Chad, Eritrea and Senegal] having 80–100% of PAs that have had no research on elephants published (Figure 3).

The biases indicate that management and conservation of African elephants informed by research may not be based on regionally accurate evidence and data. While this review focuses on savanna elephants (*Loxodonta africana*), the results seem likely to apply to other vulnerable and threatened fauna in Africa. Similar biases can be seen in research on Africa's birds where research has been heavily influenced by accessibility [57].

The geographical bias towards Kruger National Park (and surrounding reserves), South African PAs and other high profile, tourist-oriented PAs may be of concern if these parks are not representative of other areas with elephants that rely on the same research. These parks have proportionately small populations that may not be indicative of the greater elephant meta-population. These elephants may not be exhibiting standard behavioural patterns [58,59]. High profile parks draw high numbers of tourists whom observe the wildlife at close range leading to human disturbance. For example, Kruger National Park alone draws ~1.4 million tourists per annum [60]. Tourists and safari staff can cause edge effects and aggressive and defensive behaviour amongst the elephants, and their presence can override aspects of habitat selection like foraging and water access [58,59,61,62]. Although research may be needed in these areas to understand the impact of tourism and management strategies in that context, its wider applicability may be questionable if it is used to create and shape ecological management strategies across the elephants' wide geographical and international range.

Geographical biases in research may be present for a variety of reasons, but there has been no analysis of why they occur for elephants. We identify and discuss the factors behind two key obstacles that may be leading to the biases in recent elephant research: accessibility and systemic conservation research barriers.

### 4.1. Accessibility

Countries with bigger and more established research centres, such as South Africa and Kenya, are more likely to have greater capacity for maintaining long-term databases and to facilitate the processes for obtaining research permits [63]. The research centres, particularly in South Africa, are usually financially and intrinsically linked to tourism areas [64] (e.g., Kruger National Park) so roads are well established, elephant movements are known and there are safe, inexpensive places for researchers to stay [63,65]. Accessibility also includes the perceived security of researchers. Whether a country is perceived as safe may affect formal research plan approval at the researchers' home institutions and access

to research funding. This may be a particular problem with forest elephants (*Loxodonta cyclotis*) in Central Africa [50,51], and in countries that are deemed less safe due to political unrest like Zimbabwe [50,66].

Physical accessibility to elephants is a likely determinant of how researchers choose sites. For example, most of Botswana's elephants are free ranging and traverse the poorly accessible Kavango-Zambezi Transfrontier Area [3,67]. This is in stark contrast to South Africa where all elephants reside in fenced or partially fenced PAs so are mostly easily accessible [68]. These differences in accessibility could be leading to researchers preferencing easily accessed and more guaranteed data from elephants.

Politics also plays a role in the perception of security and accessibility of research. We define politics as the government's interactions with their conservation sector, particularly those involved with elephant management. For example, the last century in Zimbabwe has seen a civil war, land annexation [69], increased violence associated with illegal killing [70] and unstable politics [69] that have affected the accessibility to elephant research and the safety of researchers [70]. Most elephant research has occurred on government/state and combined private land (Figure 5). Poor security and accessibility are likely to affect the granting of research permits from home institutions, and thus limit the ability of researchers to enter countries like Zimbabwe. The difficulties of navigating the politics may be exacerbated by differences between countries. Despite 62.5% of elephants living in transfrontier areas [3], the countries involved often manage their elephant populations in different ways.

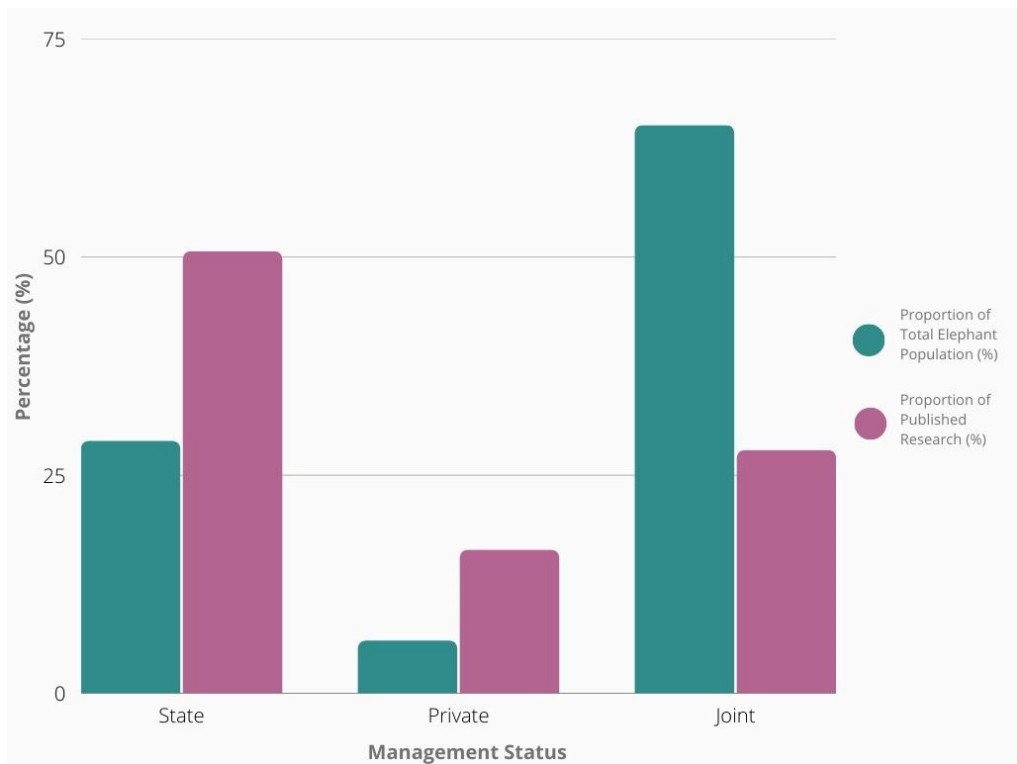

**Figure 5.** Percentage of research versus proportion of elephant population occurring in publicly managed, privately managed, or joint publicly/privately managed reserves [PAs].

Conservation research is chronically underfunded and highly competitive [47,71]. Established and well-funded research centres like Save the Elephants in Kenya, are likely to be more attractive to researchers than places without research infrastructure. The results show a bias towards these epicentres of elephant research e.g., Amboseli National Park, Kruger National Park and the Associated Private Reserves and Laipikia-Samburu Ecosystem (Figure 2). These places are relatively well funded [72], established and accessible [73]

which makes them attractive to researchers because these factors keep costs low. They are however subject to high tourism levels and the elephant populations are likely to be more disturbed. The dependency of conservation on ecotourism is a common problem and has led to the prioritisation of certain conservation areas based on tourism value as opposed to biodiversity value [74,75].

*4.2. Systemic Conservation Research Barriers*

Biases in global conservation research are becoming more apparent [76]. Our results may partially be attributed to the general bias towards the English language seen in the conservation sector [77,78]. Most countries in Africa do not use English as a first language, however, English is an official language of both Kenya and South Africa [79]. The globally used and accepted scientific process of peer-reviewed research, which has been developed and implemented by Western culture and dominated by English speaking countries [80,81], may also inhibit non-English speakers from publishing research in scientific journals. Further, research shows that countries with higher Gross Domestic Product (GDP), more English speakers and higher security have more biodiversity research being published, despite not necessarily having higher levels of biodiversity [49]. For example, South Africa is an internationally recognised biodiverse country, but so too are countries with higher elephant populations such as Mozambique and Tanzania but they are characterised by lower GDPs, fewer English speakers and more perceived issues surrounding security.

Botswana appears to be an exception to the rule and its relative exclusion from elephant research is not explained by the above arguments. Botswana is considered a safe country, has high GDP, a high proportion of English speakers, and runs well established research units like Elephants Without Borders [82]. The low level of elephant research is likely to be linked to the low accessibility of its elephants as many live in the largely inaccessible Okavango Delta and Kavango-Zambezi Transfrontier Conservation Area, as well as having more strict conservation permit and research policies than surrounding countries in southern Africa [3,67].

Solutions to help overcome biases in research are available. Extra and conscious effort is required to conduct surveys and data collection in environmentally and spatially underrepresented areas, which can be done remotely with in-country partners and remote sensing technology [46]. This study provides the necessary basis for this approach at a broad level and identifies for example a clear need for more research on elephants in Botswana and Zimbabwe, particularly in Northern Botswana and Hwange National Park. Further suggestions have been made that research should be based on environmental variables that are important to the species based on previous research and field experience, instead of based on location and accessibility [46]. In the case of elephants, this could mean targeting variables such as water availability and feeding ecology, and temporarily favouring research sites in Botswana and Zimbabwe that best represent these variables. The bias can also be countered by using the most current IUCN African elephant survey to choose survey sites based on local population density and considering levels of disturbance to populations before initiating and publishing research.

A large amount of research on elephants (>50%) has occurred on fenced or partially fenced populations compared to the number of elephants these represent (Figure 4). Therefore studies, particularly on movement ecology, may not be indicative of the greater population that are truly free ranging, for example, in transfrontier areas. While removing fences is often not feasible in the short-term, more research is needed in transfrontier areas which account for 62.5% of the total elephant meta-population [3,83,84]. As wildlife mobility and movement become more essential with changing climate and within heavily fragmented landscapes, the role and existence of transfrontier areas is gaining greater prominence [83,85]. This review shows that only 0.76% of research on elephants in the last 20 years has occurred in transfrontier parks and 4.96% of research took places across multiple meta-populations in multiple countries.

The research topics also show bias towards the areas of Ecology and Behaviour (Figure 6). While this is largely to be expected in a study focusing on ecology, the lack of research on climate interactions and climate change is of concern given the major climatic changes over prime elephant habitat in Southern and Eastern Africa during and since the Millenium Drought [7,86]. Based on ecological and behavioural studies, we know that elephants are highly susceptible to heat stress due their size, an inability to sweat and poor heat transfer mechanisms [8]. They are also highly dependent on surface water and their movements are inhibited by their proximity to a water source [8]. Under a climate change-induced drought scenario, this threat to their survival [as well as how their response may impact other species and local communities] is imminent and understudied. While behavioural and ecological studies may help inform this management issue, an explicit and evidence-based understanding will be necessary to best prepare the landscape and associated managers for the impact. This is particularly relevant for areas like Botswana, where an unexpected cyanobacterial bloom as a result of higher average ambient temperatures killed ~300 elephants [87]. It is also relevant in Zimbabwe where the drought is ongoing and severe and has caused the death of 200 elephants already [16,88]. Drought and erratic weather events in Eastern and Southern Africa are predicted to increase in occurrence, severity and longevity [7,86]. Poaching is also underrepresented in the research given it is the primary cause of population decline, though it is possible that this information is more widely disseminated through government and NGO agencies, as opposed to peer-reviewed scientific publications.

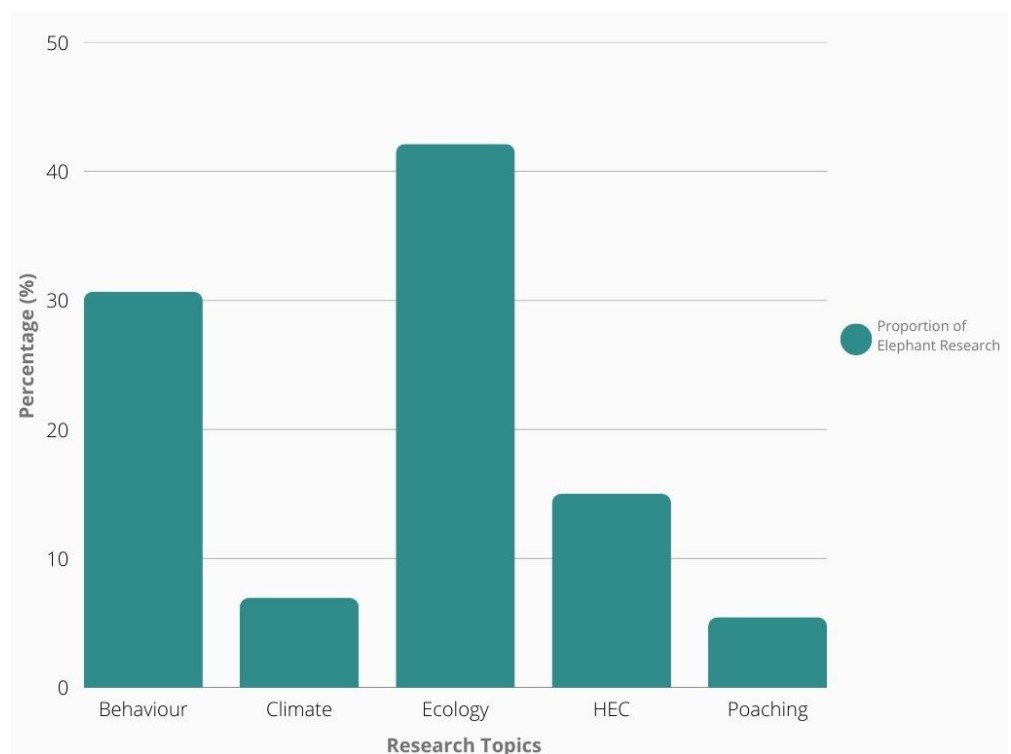

**Figure 6.** Percentage of total elephant research to which each topic contributes.

*4.3. Limitations*

This systematic review has several limitations as follows. First, we focused on ecology because of other aspects of biology and biochemistry are less likely to influence policy and management of wild elephant populations. We recognise that this may not always be the case and that there may also be biases in research areas outside of the field of ecology. Further, we have focused on literature (both as evidence and as data) on only elephants and understand that other factors may influence issues such as fencing. We believe that this was within the scope of our study on research biases but acknowledge there are further opportunities to expand and examine other research areas and topics.

We acknowledge limitations around the assumption that a study in a park represents all elephants in that area when in reality only a subset of the population may have been studied. Our aim was limited to identifying large scale biases and gaps as a starting point for examining research biases.

We also acknowledge that our omission of government reports and informal research from non-government organisations may have impacted this review. Given that many of these reports and data are not peer-reviewed, it is difficult to consider them objectively as part of this study and it is difficult to gauge how they inform elephant conservation practice. We acknowledge the likely importance of these reports, and encourage peer-review, publication, and wide-scale accessibility wherever possible.

Finally, we recognise that our assumption that elephants abide by the boundaries of countries and PAs does not reflect the reality of elephant movements. We used the population definitions provided by the IUCN elephant census and to the best of our knowledge this provides the most accurate population estimates available even if they do not necessarily capture the movements of elephant populations across borders and boundaries.

*4.4. Management Recommendations and Future Research*

This implications of this study could be discussed in various ways but we aim to present only the firmest conclusions. We recognise that it is not feasible or realistic for research to match elephant population size and only use the proportion of total elephant number as a rough guide to the extent of bias. In response we encourage researchers to consider choosing sites and populations based on the need for research as opposed to ease of access or other factors encouraging or discouraging conservation research.

We also encourage researchers to expand the topics they cover given that some research areas have become more important over time. These include human-elephant conflict and climate change which involve highly fluid issues relating to elephant management. We also recommend exploring the finer details of the administration around research permits such as cost, rules and legislation and conditions. A review of these independent variables may provide further insight into why the biases are present and help develop strategies to overcome them. Further, a lack of research in certain areas, such as climate change, may indicate they are novel areas of research and yet to reach high levels of publication.

The total African elephant meta-population has declined by 98% in the last 200 years, during European colonisation of the continent [4]. Considering that local communities have coexisted with significantly higher populations of elephants for millennia; site specific conservation research is likely to benefit from an increased role of local communities. While the human communities coexisting with elephants before colonisation were smaller, traditional knowledge and management is likely to hold great value to elephant conservation. This will help alleviate bias caused by lack of security and accessibility and make areas where research is difficult safer [89,90]. It will also provide access to a significant amount of historical data that can be drawn from local knowledge and traditional management practices. The ratio of research to elephant population is skewed on both public and jointly managed land (Figure 5), where it is evident that more research needs to occur on jointly managed land. While this may mean higher engagement of the private tourism sector, it is also an opportunity for collaborative community-based engagement and management in the future. Community based management and inclusion of local people in conservation in

Africa has led to decreased illegal killing, decreased human-wildlife conflict and increased quality of life for local people [90–93]. An interdisciplinary and socially oriented research approach will become more critical when managing landscapes, people and wildlife at any scale as the climate continues to change, communities become more empowered and traditional knowledge regains value in the conservation space.

Although we have identified geographic and population bias, we do not wish to criticise the value of research that has been conducted, or the value of well-established and long-term monitoring programs which have been pivotal to understanding elephant ecology. In particular we recognize that localised studies have great value for managing local populations (e.g., Kruger National Park). Our main recommendation is that researchers strive to overcome any inertia associated with these well-established programs and diversify according to where new research will have greatest impact. In the long term, this may lead to the establishment of more research centres and long-term monitoring opportunities as well as diversifying the evidence base available to managers and researchers.

## 5. Conclusions

The economic, cultural and ecological value of elephants is significant, and they are a high priority for conservation in Africa. This review indicates that the current literature is inherently biased towards PAs with elephant populations that are not necessarily indicative of the greater elephant meta-population. We propose that future research on elephants should be established so that these biases are acknowledged and their implications further explored and plans for pro-actively working towards overcoming bias are established. By addressing population and geographic publication biases in the evidence base for elephant research, we are likely to improve evidence-based management and conservation

**Supplementary Materials:** The following supporting information can be downloaded at: https://www.mdpi.com/article/10.3390/d15030451/s1, Table S1 Research and populations by country; Table S2 Areas with the most research published; Table S3 Areas with the highest elephant population; Table S4 Fencing status of included areas; Table S5 Management/ownership status of included areas; Table S6 Proportion of each country's elephant population that has not been researched in the last 20 years; Table S7 Count of the topic of research papers for the countries with the highest elephant popu-lations.

**Author Contributions:** R.B.G. was responsible for conceptualisation, methodology, validation, formal analysis, investigation, resources, data curation, writing (original draft), writing (reviewing and editing) and visualisation. R.H. was responsible for contributing to the above primarily through conceptualisation, methodology, supervision and writing (reviewing and editing). All authors have read and agreed to the published version of the manuscript.

**Funding:** This research received no external funding.

**Data Availability Statement:** Analysed data is available in the Supplementary Materials section of this manuscript. The full, raw dataset of reviewed literature can be provided on request as it formats poorly in a Microsoft Word document.

**Acknowledgments:** The authors would like to acknowledge the ongoing support of the Australian Government Research Training Program.

**Conflicts of Interest:** The authors declare no conflict of interest.

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
