# Peer review of "Elephants Not in the Room: Systematic Review Shows Major Geographic Publication Bias in African Elephant Ecological Research"

_diversity, doi:10.3390/d15030451_

Round 1
Reviewer 1 Report
The submitted manuscript examines research on African savanna elephants (Loxodonta africana). The assumption of the authors seems to be that the proportion of studies should reflect the geographic location and size of the populations. While an interesting premise, the idea does not reflect how science has been and continues to be done in terms of the emphasis and funding on model species (e.g., the nematode Caenorhabditis elegans, the fruit fly Drosophila melanogaster, the zebrafish Danio rerio, and the mouse Mus musculus). Similarly, the question arises as to whether what we learn about elephants in one population can be translated to other populations. We have certainly done this for a range of sub-fields within biology from reproduction to behavior. Thus, the central question becomes whether this approach is applicable to conservation and management. An important dichotomy exists here. The basic science may translate very well from one population to another. For example, reproductive aspects, sensory biology, and physiology. However, management decisions may require more population specific data to make informed decisions, and we do know that basic biology can be affected by climatic, habitat, and anthropogenic factors.
Introduction
While the abstract focuses on where research has been conducted, the introduction covers a more expansive set of issues that the authors relate to where elephants have been studied: Protected Areas, Fencing, Carry Capacity (whether in fenced areas or not), Impacts of Elephants on Vegetation and Other Species, Climate Change, and the Economic / Ecological Implications of Elephant Conservation. The writing is very good, but the logic of the order of presentation is lacking. Are PA’s the biggest issue and fences the greatest concern (there is a great deal of text on these two items)? Paragraphs 2-5 not only raise issues but include points of view by the way in which they are written. At the end of the first paragraph, the authors mention major issues of “illegal ivory trade and conflict with humans” as well as “Climatic changes,” but the ensuing paragraphs provide relatively little elucidation of these issues relative to elephant distribution. If the introduction is issue-focused on conservation, why are the topics for the following paragraphs not organized as shown in Table 1 (HEC, Ecology, Behaviour, Climate, Poaching)? Instead, the authors delve into protected areas and fencing.
In this regard, the authors make an apparent connection that is erroneous. In the second paragraph of the introduction they cite Chase et al. (2016) who state that 84% of African savanna elephants reside within Protected Areas (PAs). While the continental wide survey done by Chase et al. had some limitations, this statement was made in their publication. In the next line of the manuscript’s introduction, the authors write “Fence areas generally ….” No where in the Chase et al. paper is the word “fence” used. A PA should not be equated with a fenced area. The linking of these two sentences makes that apparent connection. This needs to be rewritten (and clearly the authors recognize this per the presentation of their data, so it is just an issue of the juxtaposition of the statements not the knowledge of the authors). The authors could provide what percentage of protected areas that have elephants are fenced at this point to help clarify.
[Also, the recommendation and action have been to remove fences more than increase them (Linden et al. 2022. Ecosystems). The authors might consider reviewing the recent literature on the role of fencing in management in sub-Saharan Africa beyond papers that contain the key word elephants to provide this broader perspective.]
Overall, paragraphs 2-4 do a rather uneven job of setting up the motivation for the study. They are more about justifying the need for elephant conservation yet criticizing current strategies apparently because the right populations of elephants have not been studied although this is not stated and one might think it would be given the wording of the abstract. A more informative approach would to detail how the problems facing elephant conservation (e.g., Topics in Table 1) differ by region (e.g., the Sudan and South Africa are both geographically and conservation issue wise quite distant). In addition, the economics and governmental stability of countries with elephants are not discussed and are not provided in Table 1, yet political stability seems a prominent finding (section PAs with and without research).
I concur that a hypothesis-driven approach is valuable, but the first four of the hypotheses couple a prediction with a singular cause in each case. Thus, the authors must test not only is there a geographic bias (H1) but that the reason is because these countries have established elephant research units / NGOs (South Africa and Botswana). A host of other factors could correlate with the one they select in each case (H2: pre-existing biases in global conservation research; H3: higher funding and accessibility; H5: these areas are easier to access). H4 seems to be a rewording of H1. The explanations for the hypotheses here suggest that if the hypothesis is supported than the cause also is understood but that is an inappropriate interpretive leap.
A final note on the introduction: In the methods paragraph just below Table 1, the authors write “We limited the search to the last 20 years because the main interest was understanding how elephants need to be managed under the effect of climate change ….” I did not obtain this primary interest from the introduction.
Methods
The authors used the” number of papers published per country and per area/reserve” as their means of determining research effort. The assumption here is that elephants respect country boundaries, which of course they do not. This also is an assumption of the IUCN report that the authors used, so unless they were to examine the ranges of populations and adjust by the proportion of each within a country, they are left with the data available. However, they should discuss this methodological issue. In addition, some countries have work performed by government agencies (not unlike the US for example) in which publications are reports and not peer-reviewed in the primary literature. These reports inform management practices, but they are not likely to have been picked up in the literature review process.
It is difficult to evaluate how useful the “partially fenced” designation is for research and management. What is the operational effect on elephant movements of partial fencing (e.g., Table 1, Figure 4).
Results
As stated by the authors, the significant different in distribution of papers published compared to elephant population sizes is because of two countries, Kenya and South Africa. Surprisingly, Figure 1 is not that far from a true correlation between the two variables. What would the South Africa papers published bar look like if you removed studies conducted on smaller, fenced populations (given that all populations in this country are fenced or were fenced)?
In the section “Fencing and custodianship of elephant habitats” please put the order of the research and the residence of elephants in the same order (e.g., unfenced, partially fenced, and fenced) rather than the first in this order and second in the reverse order.
Research Topics: It would be nice to see a graph with time as the x-axis and different lines for the topics given that HEC and climate change have been increasing in impact and importance for research over the past two decades.
**The text below Figure 6 appears to be left from the directions** “This section ….”
Discussion
The authors make an assumption that is questionable on the relationship between studying (or not) a particular population and the ability to manage elephants. Scientific research is based on appropriate sampling of populations and generalizing the findings. The third paragraph of the Discussion “The biases indicate…” that current practices require regionally accurate evidence and data. At some level, this is clearly impractical. If we have to test whether findings from one study require verification in every location to which they are applied, then we have lost the generality of science. Clearly, cases exist where such research is absolutely necessary because the local conditions are different from where studies were conducted, but in many other situations, this would not be the case. Depending on the situation, action based on the best available information is better than waiting and gathering data on the local conditions. Resource availability along with other factors have to weighed, and the authors should consider expanding this paragraph to consider the costs and benefits of the range of potential actions.
South Africa is an anomaly in many ways and much of the research in KNP and the Associated Private Nature Reserves is focused on management of their own populations, and beyond work on the basic biology of elephants, other countries clearly recognize the difference in policies on the ownership of wildlife in South Africa and the strong management philosophies. The citation of the Szott et al. 2019 (59) paper here is interesting as these authors state, “We present the first study investigating the impact of wildlife tourism on African elephant, Loxodonta africana, behaviour.” If indeed only one such study has been done, I do not believe that other locations would apply such findings without further consideration. I do not disagree with what the authors write, but it seems a rather flimsy argument. Through this discussion, the authors do raise another point for analysis and that is the degree of tourism by country and how that relates to both the number of publications and they type of research conducted.
Accessibility
This section is fine, except that it would have been nice to analyze results in this regard or to show a graph correlating accessibility with other independent variables. For example, factors such as the costs of permits, days from permit application to acceptance, rules for who is permitted to conduct research in a country, stipulations on the presence of permanent camps and government personnel (e.g., wildlife officers). I would leave forest elephants out of the discussion given that you did not include them in the analyses. Another factor that you omit (or I missed) is the benefit of long-term studies of a population, such as working with known elephants with tracked demographics, and what can be learned from adding to the knowledge base rather than starting anew elsewhere.
Systemic conservation research barriers
The paragraph “The large and …” is a bit self-aggrandizing. The current manuscript reveals a discrepancy between elephant distributions and publications. The manuscript does not analyze the type of research conducted and how it is being used across populations for conservation. The authors presume that work done in one locale is not appropriate for informing elephant management in a different location. No evidence is provided to support this contention.
Similarly, the statement “a disproportionate amount …” is value statement. The paper does not examine why this is inappropriate nor how the data from these studies have been extrapolated inappropriately and harmed the management of elephants elsewhere. The authors are going well beyond their findings to make such statements.
The next paragraph, “The research topics,” uses the word “bias” and this word carries with it the idea of prejudice or a conscious favoritism (although fortunately in society the idea of unconscious bias is being recognized, studied, and discussed). Perhaps a word such as “emphasis” would be more appropriate. The word “bias” has strong connotations and the tone of the paper would be improved if its use were curtailed (e.g., see conclusions).
Bias – “implies an unreasoned and unfair distortion of judgment in favor of or against a person or thing” (Merriam-Webster).
In terms of climate work, a graph on paper topics over time would be informative (see recommendation above). It is also possible that such work is being done more broadly such that elephants would not be the focus yet such work can still inform management at the landscape scale.
Limitations
I agree with the limitations stated. Unfortunately, the viewpoints expressed previous to this section have created a tone for the paper that cannot be undone by this section. Again, I am not clear as to the “purpose of this paper” as it seems much more than to show a disjunct between the distribution of elephant populations and papers published. Nowhere in the paper do the authors provide data that supports their contention that “the current literature in inherently biased towards PAs with elephant populations that are not necessarily indicative of the greater elephant meta-population.”
As a final question to the authors, who should control the body of research conducted on elephants such that the distribution of work reflects the distribution of elephants?
Overall, the authors present some interesting data. The introduction should be rewritten to align better with the hypotheses (which also should be reconsidered, see above). Some additional analyses and presentation of data would be interesting. The discussion is overly long and goes well beyond the data. The word “bias” should be dropped. The limitations should be more evident from the start of the paper and the applications of the findings be framed within these limitations.
Reviewer 2 Report
Thank you for the opportunity to review this manuscript. It was an interesting and timely read, especially as we must prioritize resources to addresses conservation challenges facing this endangered species. Before acceptance for publication, I have several questions that I'd like the authors to consider, along with a few suggested revisions.
First, your hypotheses as you present them at the end of the Introduction seem to overlap too much, and for at least one you don't provide evidence to support/refute them. For example, Hypotheses 1 and 3, and Hypotheses 2 and 4, seem like they could be combined. You mention GDP in the second hypothesis, but this is never analyzed in your results. Please revise and streamline these hypotheses in a way that reflects how you analyzed the data.
When classifying the research topic for each study, were these topics mutually exclusive? In other words, could a study include two or more topics? It's not clear in the methods section, and from Figure 6, it appears that they were mutually exclusive. I would think that many studies cover multiple topics, so this part of the analysis may not accurately reflect patterns in the literature.
Next, how did you deal with the possibility that elephants may move across political borders? What about the potential effects of spatial autocorrelation?
Maybe this is my most important question: why do you imply population size the only (or most relevant) factor to determine how much need a population has for research attention? Surely, some areas of Africa have relatively little research compared to elephant population size, but these areas may also have a suitable number of elephants for that ecosystem. Similarly, countries with disproportionately high rates of research may be more deserving, both practically and ecologically (e.g., Kenya had an elephant population crash that is slowly recovering, offering a lot of insight into similar events that may happen elsewhere). The reasoning behind your analysis seems to rest on this assumption, but I don't know if it's necessarily the right one.
In the third paragraph of subheading "Systemic conservation research barriers" in the Discussion: you state that variable such as water availability and feeding ecology should be preferentially studied in places without much research. If careful study of these factors is required, why wouldn't it be appropriate to document the influence of these factors on elephant ecology in places with good research infrastructure (e.g., South Africa, Kenya)? Would you expect elephant populations in these ecosystems to respond differently compared to other locations? I'm just not sure that it's worth putting in resources to collect lower quality data in places with less infrastructure, especially if the patterns we see in response to these factors would be similar/identical.
In the third sentence in the subheading "Limitations" in the Discussion: I'm not sure a personal belief is warranted here, as there is ample evidence that these studies (e.g., on captive animals) help inform management practices for wild elephants (see Bechert et al., 2019).
Minor comments/revisions:
Throughout manuscript, make sure to italicize scientific names (e.g., Loxodonta africana and Loxodonta cyclotis). You can also abbreviate genus after first presentation.
Second sentence of Introduction: do elephants only maintain and shape ecosystems across southern Africa? (I think "no.")
First paragraph of Introduction: what's the source for 98% population decline?
Second paragraph of Introduction and throughout manuscript: I believe the more appropriate/politically correct term for poaching is now "illegal killing." Please consider changing.
Sixth paragraph of Introduction, first sentence: Is reference 46 really relevant here?
First paragraph of Methods: it looks like a few of your search terms are exact duplicates of each other. Please review carefully and change as appropriate.
First paragraph, second sentence in subheading "Data analysis" of Methods section: what does "no notable discrepancies" mean?
After Figure 6, remove text leftover from manuscript template.
Fill in appropriate information in end text (e.g. "Supplementary Materials", remove template text after Appendix A header).
Reviewer 3 Report
The authors are praised for producing this submission as a systematic review of this needed topic. The inclusion of their professional opinions about the African elephant is appreciated but the opinions can be linked to plan of action as recommendations. The authors may insert a set of recommendations using the previous published literature as evidence. The submission then can have more impact on the direction of the ecological research. However this is only a suggestion and should not be considered as a condition for accepting the publication.
Round 2
Reviewer 1 Report
I do not feel that the authors made a concerted effort to address the extensive comments that I made on the first submission. I find many of the same flaws in this version.
Reviewer 2 Report
Thank you to the authors for carefully addressing my previous comments. I have one remaining concern that needs to be addressed before publication. Because now I know that the research topics were not mutually exclusive, I'm not sure that the percentages provided in the paper are correct, as the percentages add up to ~100% (100.1% by my simple addition). Can you please double-check and confirm, as I would expect many studies to overlap several research topics, and it doesn't seem like hardly (if any) did in your calculations. Further, because topics are non-mutually exclusive, it's not appropriate to use a pie chart (Figure 6) to depict the results--these charts are typically useful only when percentages are supposed to add to 100%. I recommend changing to a bar chart.
Round 3
Reviewer 2 Report
Thank you for revising the manuscript. I believe it is now suitable for publication.